# Automated Structural Analysis and Quantitative Characterization of Scar Tissue Using Machine Learning

**DOI:** 10.3390/diagnostics12020534

**Published:** 2022-02-19

**Authors:** Luluil Maknuna, Hyeonsoo Kim, Yeachan Lee, Yoonjin Choi, Hyunjung Kim, Myunggi Yi, Hyun Wook Kang

**Affiliations:** 1Industry 4.0 Convergence Bionics Engineering and Marine-Integrated Biomedical Technology Center, Pukyong National University, Busan 48513, Korea; maknuna@pukyong.ac.kr (L.M.); hyeonsookim@pukyong.ac.kr (H.K.); yclee@pukyong.ac.kr (Y.L.); 2Department of Biomedical Engineering, Pukyong National University, Busan 48513, Korea; yoonjin_@pukyong.ac.kr (Y.C.); guswjd3486@pukyong.ac.kr (H.K.)

**Keywords:** scar tissue, machine learning, structural analysis, pathological features, histological analysis

## Abstract

An analysis of scar tissue is necessary to understand the pathological tissue conditions during or after the wound healing process. Hematoxylin and eosin (HE) staining has conventionally been applied to understand the morphology of scar tissue. However, the scar lesions cannot be analyzed from a whole slide image. The current study aimed to develop a method for the rapid and automatic characterization of scar lesions in HE-stained scar tissues using a supervised and unsupervised learning algorithm. The supervised learning used a Mask region-based convolutional neural network (RCNN) to train a pattern from a data representation using MMDetection tools. The K-means algorithm characterized the HE-stained tissue and extracted the main features, such as the collagen density and directional variance of the collagen. The Mask RCNN model effectively predicted scar images using various backbone networks (e.g., ResNet50, ResNet101, ResNeSt50, and ResNeSt101) with high accuracy. The K-means clustering method successfully characterized the HE-stained tissue by separating the main features in terms of the collagen fiber and dermal mature components, namely, the glands, hair follicles, and nuclei. A quantitative analysis of the scar tissue in terms of the collagen density and directional variance of the collagen confirmed 50% differences between the normal and scar tissues. The proposed methods were utilized to characterize the pathological features of scar tissue for an objective histological analysis. The trained model is time-efficient when used for detection in place of a manual analysis. Machine learning-assisted analysis is expected to aid in understanding scar conditions, and to help establish an optimal treatment plan.

## 1. Introduction

Scar tissue develops as a result of tissue damage caused by injury, surgery, or burns. Inflammation, proliferation, and remodeling are the three stages of the healing process [1]. Scars can be faded or removed, using a variety of techniques, including corticosteroid injections [2], cryotherapy [3], and laser therapy [4]. Identifying and evaluating scar tissue is the most crucial step in determining the extent of tissue damage and planning a suitable treatment pathway for scar management and removal [5]. However, determining and analyzing the scar regions in the skin tissue remains challenging. The limited field of view often restricts analysis of the entire scar lesion in the whole slide image (WSI). For instance, recent research studies have simply manually located scar lesions to analyze a region of interest (ROI) in the scar area [6,7]. An in-depth analysis of a scar region is also difficult, owing to the comparable color distributions in hematoxylin and eosin (HE)-stained images. Although morphological changes in the scar region can be identified and distinguished, histological analyses often suffer from ambiguous classifications of tissue components in the scar region [8,9]. Additionally, HE-stained image suffers from high memory usage and is often labor-intensive in manual analysis.

The orientation of collagen fibers is an important factor in distinguishing the tissue characteristics between normal and scar tissue [10,11]. Several studies have used second harmonic generation (SHG) microscopy to examine collagen fiber formation [12,13]. Bayan et al. [14] used SHG imaging combined with the Hough transform to characterize the important features and orientation of collagen fibers in collagen gels for disease diagnosis. Histological analysis has also been used to visualize and analyze collagen fiber formation [15]. Several histological dyes, including HE [16], picrosirius red [17] and Masson’s trichrome (MT) [18] have been used to stain tissue components, to improve tissue contrast, and to highlight cellular features for in-depth analysis. MT is one of the most commonly used staining dyes in histological images for identifying the distribution of collagen [19]. MT stains collagen fibers in blue using three staining colors, so as to distinguish them from the other microstructures [20]. Tri Tram et al. used a convolutional neural network (CNN) model to classify both normal and scar tissues, as well as to visually characterize collagen fiber microstructures based on features extracted from the developed model, such as the collagen density and directional variance (DV) [21].

Machine learning and deep learning techniques have been used to analyze medical images from X-rays, computational tomography, ultrasound, and magnetic resonance imaging, and have shown high accuracy and reliability [22]. Machine learning techniques can help increase performance (i.e., speed, automation, and accuracy) and provide reliable results from iterative calculations, making machine learning analysis more effective than conventional methods (Figure 1) [23]. Unsupervised machine learning, such as K-means, is used to segment and cluster unlabeled data [24] by learning the high-level features from convolutional and pooling processes. The CNN has become the most popular deep learning model, as it is capable of recognizing key features [25]. CNN methods have been widely applied in the classification and segmentation of a vast number of medical images [26]. The class activation mapping method is useful to understand the predictions by mapping important features based on the convolution output. Transfer learning and progressive resizing methods could also increase the performance of CNN model training [27].

In view of the above, in this study, machine and deep learning techniques based on object segmentation were developed, aiming to effectively classify and characterize scar lesions in a WSI with a Mask region-based CNN (Mask-RCNN) and object segmentation, and to overcome the limitations in current histology analysis. The Mask RCNN was initially developed from the Faster RCNN, which generates classification, box regression, and branch additions to predict a mask area for each ROI [28]. The feature pyramid network (FPN) has also been used to increase the accuracy and speed of the analysis [29].

## 2. Materials and Methods

### 2.1. In Vivo Scar Model

Four male Sprague Dawley rats (7 weeks; 200–250 g) were raised in a controlled room (temperature = 25 ± 2 °C and relative humidity = 40–70%) with an alternating 12 h light/dark (wake-sleep) cycle (i.e., on at 7 a.m. and off at 7 p.m.). During the tests, all animals were under respiratory anesthesia using a vaporizer system (Classic T3, SurgiVet, Waukesha, WI, USA). Initially, 3% isoflurane (Terrell™ isoflurane, Piramal Critical Care, Bethlehem, PA, USA) in 1 L/min oxygen was delivered into an anesthesia induction chamber, and 1.5% of isoflurane was supplied to maintain anesthesia via a nosecone. Then, an electric hair clipper and waxing cream (Nair Sensitive Hair Removal Cream, Nair, Australia) were used to completely remove the hair on the buttocks of the anesthetized animals for maximum light absorption by the skin. All animal experiments were approved by the Institutional Animal Care and Use Committee of Pukyong National University (Number: PKNUIACUC-2019-31). The current study used laser-induced thermal coagulation to fabricate a reliable mature scar model on the animal skin, according to previous research [30]. A 1470-nm wavelength laser system (FC-W-1470, CNI Optoelectronics Tech. Co., China) was employed in a continuous-wave mode to induce the thermal wound, owing to the strong water absorption (absorption coefficient = 28.4 cm${}^{-1}$ at 1470 nm) and short optical penetration depth in the skin. A 600-µm flat optical fiber was used to deliver the laser light to the target tissue. Before laser irradiation, a laser power thermal sensor (L50(150)A-BB-35, Ophir, Jerusalem, Israel) in conjunction with a power meter (Nova II, Ophir, Jerusalem, Israel) was used to measure the output power of the optical fiber. For testing, the flat fiber was situated 25 mm above the skin surface, and the beam spot size on the surface was 0.3 cm${}^{2}$. The targeted tissue was irradiated at 5 W (power density = 16.7 W/cm${}^{2}$) for 30 s (energy density = 500 J/cm${}^{2}$) to generate a circular thermal wound with minimal or no carbonization. As a result, a 1–2 mm thick section of coagulated tissue, 10 mm in diameter, was created in the epidermis and dermis on each side of the animal buttocks (N = 8). Four weeks after irradiation, the thermal wound became dense scar tissue via wound healing.

### 2.2. Histology Preparation

All of the tested scar tissues were harvested from animals 30 days after laser irradiation. The extracted scar tissues included both the scar tissue and the surrounding normal tissue. After tissue harvesting, all samples were fixed in 10% neutral buffered formalin (Sigma Aldrich, St. Louis, MO, USA) for two days. The fixed samples were dehydrated, cleared, and infiltrated sequentially using an automatic tissue processor (Leica TP1020; Leica, Wetzlar, Germany). Paraffin blocks were fabricated and divided into slices with a thickness of 5 µm to prepare histology slides. The histology slides were serially sectioned (N = 6 slides per block; total of 48 slides) at 50-µm intervals to monitor morphological variations. All of the histological slides were stained with HE (American MasterTech, Lodi, CA, USA) to qualitatively assess the morphological changes and visualize the collagen fiber distribution in the tissue. A Motic digital slide assistant system (Richmond, British Columbia, Canada; 40X and 0.26 µm/pixel resolution) was employed to capture microscopic images of the HE-stained histology slides (Figure 2) and to prepare the datasets for machine learning.

### 2.3. Scar Recognition: Mask Region-Based Convolutional Neural Network (RCNN)

In this study, the Mask RCNN was selected for object detection and instance segmentation to recognize the scar in the WSIs of the HE-stained tissues. Mask RCNN provides classification, localization, and mask prediction, thereby providing an essential benchmark for object detection [31]. The main idea is to label each pixel corresponding to each detected object by adding a parallel mask branch [32]. The Mask RCNN was adapted from MMDetection tools, a toolset comprising numerous methods for object detection [33]. In spite of its specific detection methods, the model architectures in MMDetection have typical components, such as a convolutional backbone, neck, and head. Here, the convolutional backbone was used to extract the features from the entire image [34], and four backbones were selected to extract the features from the WSI: ResNet 50, ResNet 101, ResNeSt 50, and ResNeSt 101. Next, a region proposal network (RPN) was used as a neck stage to provide a sliding window class-agnostic object detector [35]. The RPN was developed to predict both the bounding box and class labels from the extracted features [36]. Object detection models could fail to detect when facing the varying sizes of the objects that have low resolution. RPN in the Mask R-CNN uses multi-scale anchor box to enhance the detection accuracy by extracting features at the multiple convolution levels of the object [37,38]. The ROI head generated mask predictions, classifications, and bounding box predictions. The entire process is shown in the Mask RCNN diagram in Figure 3.

### 2.4. Machine Learning

Obtaining medical data is difficult and causes a problem of small sample size. Data augmentation has been addressed to generate more samples. Image manipulation and Generative adversarial network model [39] can be used. Here, image manipulation of pure rotation was used to generate 372 whole slide images, from originally 93 slide images. They were randomly selected to train (239 images, 64%), validate (59 images, 16%), and test (74 images, 20%) the neural network model. Train-validate-test split was used to validate the deep learning method. Train set was used for training the model, validate set was used for justifying the performance of the model during training process, and test set was used for the final validation. As the dataset had images of various sizes, the Mask RCNN model automatically resized the maximum scale to 1333 × 800 [40]. The scar detection Mask RCNN model was trained through the entire set of stages for 600 epochs, and was implemented in Python with Caffe frameworks. The use of fine-tuning affected the computational time and cost memory in each epoch. The fine-tuning employed an adjustment parameter to enhance the effectiveness of the training process, and was repeated frequently to increase the accuracy of the model [41,42], as shown in Table 1. After the training was completed, checkpoint files were obtained containing the Mask RCNN model parameters for the target detection and identification. A collection of scar data was then employed in the Mask RCNN testing program to obtain the trained model parameters. Regarding the implementation details, a GPU NVIDIA GeFORCE RTX 2080 Ti was used for the entire process. Torchvision 0.7.0, openCv 4.5.2, mmcv 1.3.5, and MMDetection 2.13.0 were installed as the environments for all models.

### 2.5. Evaluation Metrics

The evaluation metrics were based on using the MMDetection tools to obtain the precision, recall, loss, accuracy, and confidence score. In general, the multi-task loss function (L) of a Mask RCNN combines the losses of the classification ($L_{cls}$), localization ($L_{box}$), and segmentation mask ($L_{mask}$), as follows:(1)$$L=L_{\mathrm{cls}\;}+L_{\mathrm{box}\;}+L_{\mathrm{mask}\;}$$

The loss function for classification and localization ($L_{cls+box}$) is defined as follows:(2)$$L_{\mathrm{clstbox}\;}=\frac{1}{N_{\mathrm{cls}\;}}\sum_{i}L_{\mathrm{cls}\;}\left(p_{i},p_{i}^{*}\right)+\frac{\lambda }{N_{\mathrm{box}\;}}\sum_{i}p_{i}^{*}\cdot L_{\mathrm{reg}\;}\left(t_{i},t_{i}^{*}\right)$$

In the above, *i* is the index of an anchor in the mini-batch, $p_{i},p_{i}^{*}$ are the predicted probability and ground-truth label, respectively, $t_{i},t_{i}^{*}$ are vectors representing the four coordinates and ground truth box associated with the positive anchor, and λ is a balancing parameter for normalization. The loss function from the model attempted to learn a mask for each class, and no competition occurred among the classes for generating the masks. The mask loss was defined as the average binary cross-entropy loss, as follows:(3)$$L_{\mathrm{mask}\;}=-\frac{1}{m^{2}}\sum_{1\leq i,j\leq m}\left[y_{ij}\log{\hat{y}}_{ij}^{k}+\left(1-y_{ij}\right)\log\left(1-{\hat{y}}_{ij}^{k}\right)\right]$$

Here, $y_{ij}$ is the label of a cell (i,j) in the true mask for the region of size m×m, and ${\hat{y}}_{ij}^{k}$ is the predicted value of the same cell in the mask learned for the ground truth class *k* [43]. In addition to obtaining the confidence score, the SoftMax classifier (σ) converted the score from the SoftMax calculation into probabilities [44]. All of the evaluation metrics were computed using MMDetection tools [33], and were adapted to a scar detection artificial intelligence (AI) model.

### 2.6. Scar Extraction

To evaluate the performance of the scar mask prediction, the scar area measurements from the conventional method and machine learning were compared. In the conventional method, the WSI was cropped, and non-scar areas were selectively removed. Image J (National Institute of Health, Bethesda, MD, USA) was used to estimate the scar area by obtaining the scar pixel values [45]. The scar measurements for mask prediction were automatically calculated by using the proposed model for each backbone. A statistical analysis using the analysis of variance (ANOVA) was performed to determine the statistical significance between the conventional method and proposed method.

### 2.7. Tissue Segmentation: K-Means Clustering

The HE-stained tissue images were segmented to distinguish and clarify the morphologies of the normal and scar tissues. Tissue segmentation using K-means clustering was deployed to cluster each color point from each tissue image, and then to segment the main features, such as the collagen, hair follicle (HFs), glands (Gs), and nuclei (N) from each image. ROI inputs of various sizes were tested, i.e., 886 × 1614, 750 × 500, 500 × 500, and 250 × 250 pixels. Figure 4 describes the entire scar characterization process. The first step comprised the transformation of the tissue color space (Figure 4a). As the ROI input had an RGB color space that was unstable in terms of chrominance and luminance, the CIE L*a*b color space was chosen, and the main features were brought from the RGB color space to the CIE L*a*b as a stable color space [46]. All of the color information was in the ‘a*’ and ‘b*’ layers [47], and L was used to adjust the lightness and darkness of the image. After the color space transformation, the K-means clustering algorithm separated each data point from the tissue image into three groups: collagen area, foreground, and background. These groups were labeled with a number (0, 1, and 2) using the algorithm. The algorithm measured the distance between each cluster and the three centroids one-by-one. The algorithm then grouped the data points with the closest centroids. After grouping, the K-means clustering algorithm provided the collagen area segmentation (CAS), foreground segmentation (FS), and background segmentation.

### 2.8. Collagen Density and Directional Variance of Collagen

The CAS, as characterized by K-means clustering, was selected as a feature for generating a collagen density map (CDM) (Figure 4b). A collagen mask was created by masking the collagen-positive pixels, and was convolved with an airy disk kernel to produce a map of the fiber density (m) [48]. Moreover, a vector summation method [49] was adopted to calculate the fiber orientation (Figure 4c). The developed algorithm defined the fiber orientation by identifying the variability of the image intensity along the different directions surrounding each pixel within an image. All of the positive pixels passing through the center pixel and the angles associated with these orientation vectors were calculated. After the X and Y components of the orientation were acquired, a spatial convolution using the airy disk kernel was conducted to obtain the vector summation and determine the magnitude of the resultant orientation (R). Lastly, normalization was performed attain the DV of the collagen, as follows:(4)V(x,y)=1−R(x,y)/m(x,y)

Here, *V*, *R*, and *m* denote the DV, magnitude of the resultant orientation, and CDM, respectively.

### 2.9. Statistical Analysis

Statistical analysis All of data were expressed as mean ± standard deviation. The data were analyzed using Student’s t-test for two-group comparisons, or a one-way ANOVA for multiple-group comparisons. The statistical significance was set at *p* < 0.05.

## 3. Results

### 3.1. Scar Recognition

Figure 5 shows a scar identification and comparison of scar areas from using the various Mask RCNN backbones. Figure 5a demonstrates a WSI of HE-stained tissue with the scar prediction highlighted in a black box. The annotated and predicted scar masks are depicted with blue and yellow colors, respectively. Each image has a class label, and a confidence score in each scar bounding box. The function of the confidence score is to eliminate the false positive detection of the bounding box [50]. The confidence score in this model ranges from 0 to 1, and the dataset has a higher value (1) in each bounding box. A higher confidence score denotes a more appropriate AI model for predicting the scar area in the WSI. However, over confidence can occur owing to the class prediction for scar and non-scar areas, which are often rare and ambiguous classes [44]. Accordingly, a threshold provided a good balance of a high detection rate with few false positives, and was set automatically from the MMDetection tools (0.5).

Figure 5b shows the four backbones, along with the scar masks and bounding box predictions. Regardless of the backbones, the Mask RCNN successfully recognizes the scar area in the WSI of the HE-stained tissue. ResNet 101 outperforms the other backbones in terms of mask branch, and the bounding box prediction fits into the annotation data (as ground truth of this model) Table 2. Figure 5c compares the measured scar areas between the conventional and machine-learning methods. The measurement is based on the calculations of the respective total pixel areas in the scar lesions for each backbone. The statistical analysis of the scar measurement confirms that ResNet 101 is slightly higher than the other backbones. The machine learning shows a consistent value for each measurement, depending on the standard deviations from the measurements. Meanwhile, the conventional method even includes non-scar regions, thereby increasing the mean and standard deviation values compared to the machine learning method. Furthermore, regarding the evaluation metrics, such as the precision and recall in the mask and prediction box, ResNet 101 shows higher values than the other metrics, implying that the performance of ResNet 101 is the closest to the total pixel value for the annotated scar in predicting the mask and bounding box. The mean average precision and mean average recall (mAR) from the mask prediction and bounding box confirm the results of the ResNet 101 visualization, as shown in Figure 5b.

The computational time for predicting the scar area in the tissue was also acquired for the total number of test images (74 images) Table 2. ResNet 50 is superior in terms of time performance, ResNet 101 and ResNeSt 50 have comparable times, and ResNeSt 101 is relatively slower. Table 2 summarizes the results of the evaluation metrics for all of the backbone networks.

### 3.2. Scar Characterization

Figure 6 shows a characterization of a scar in the HE-stained tissue. Figure 6a illustrates the dermal regions acquired from the WSI with a scar region highlighted by a black box. Markedly, the scare region is full of dense fibrotic collagen, in an organized manner. Figure 6b presents a CDM corresponding to the WSI in Figure 6a. The scar region is evidently identified in a red color (yellow dashed box) to highlight its higher collagen density, relative to the surrounding normal tissue. Figure 6c shows a DV map for indicating the orientation(s) of the collagen fibers. Lower variance values indicate that the collagen fibers are highly oriented (directional). Similar to Figure 6b, the scar region shows a lower variance, owing to the abundant presence of the collagen.

Figure 7 demonstrates a tissue characterization of three regions (normal, mixed, and scar) using k-means clustering-based color segmentation. The mixed region indicates a mixture of normal (right) and scar (left) tissues. Figure 7a represents the original input from the three regions with a size of 886 × 1614 pixels (acquired from HE-stained tissue). Figure 7b presents the results of the FS on the WSI from Figure 7a. The K-means clustering vividly segments the HFs, Gs, and N in purple. The absence of HFs and Gs in the scar tissue indicate irreversible thermal injury, as well as an excessive formation of collagen in the tissue. However, the mixed region (middle) contains HFs, Gs, and N in the normal region (non-scar) but no HFs, Gs, and N in the scar area in the same ROI. Both the CAS and CDM show that the normal tissue is associated with more black spots, representing HFs, Gs, and N (Figure 7c,d. In contrast, the scar tissue shows no or minimal black areas, indicative of the presence of dense and excessive collagen. Figure 7e shows the percentage of pixel values from each segmentation and DCM. The FS shows that the normal tissue has a greater number of HFs, Gs, and N than the scar and mixed tissues (*p* < 0.05). The difference between the normal and scar tissues in the FS is insignificant, because the scar tissue still has N distributed throughout the entire tissue. Both CAS and CDM show that the scar area has more collagen than the normal and mixed tissues (*p* < 0.005).

Figure 8 compares the scar characterizations of three different ROI sizes (250 × 250, 500 × 500, and 750 × 500) in terms of the FS, CAS, CDM, and DV, so as to validate the applicability of the developed algorithm to various image sizes. Regardless of the image size, the characterization algorithm identifies the excessive presence of densely oriented collagen without HFs, Gs, and N in the scar areas.

The images in Figure 8 and Figure 9 quantify the extent of the FS (Figure 9a), CAS using K-means clustering (Figure 9b, CDM (Figure 9c), and DV (Figure 9d). Regardless of size, the FS shows that scar tissue has fewer pixel areas representing HF, G, and N (50%) than normal tissue (88%; *p* < 0.001 for 750 × 500; Figure 9a). The reduction occurs owing to the absence of HF and G in the scar tissue resulting from the irreversible thermal injury. According to the CAS (Figure 9b), the scar tissue has larger collagen areas (up to 92%; magenta color in Figure 8c) than the normal tissue (58%; *p* < 0.001) for all sizes, representing the collagen expansion owing to fibrotic activity from tissue injury. The CDM demonstrates that the scar tissue has a two-fold higher collagen density than the normal tissue (96% for scar vs. 48% for normal; *p* < 0.001), irrespective of the image size. However, the DV shows that normal tissue is associated with higher variances than scar tissue (96% for normal vs. 52% for scar; *p* < 0.001). Both the CDM and DV validate that upon tissue injury, the collagen formation in the scar tissue is densely oriented in a relatively organized manner.

## 4. Discussion

In the current study, the AI models trained by the Mask RCNN show outstanding performance for scar recognition in various unstructured sizes of HE-stained tissues. A previous study classified normal and scar tissues by using the modified “VGG” model for an ROI image of MT-stained tissue, as commonly used for collagen extraction [21]. In this study, the Mask RCNN was able to classify and localize the scar area in an WSI under challenging conditions, such as with a limited size of the input in the Mask RCNN (1333 × 800 pixels) [51], and unspecified staining for the collagen extraction [52]. Hence, the Mask RCNN attained the best results for scar recognition in HE-stained tissues, with high accuracy. The features extracted from the scar regions depended on the backbone performance. The combination of both the backbone and RPN aided in increasing the performance in terms of the feature alignment and computational time [53]. All backbones showed poor performance in predicting scar lesions in the WSI using the Mask RCNN method. However, ResNet 101 had a slightly better result (depending on the evaluation metrics) than the other backbones for scar lesion prediction. The advantage of ResNet is that the performance of this model does not reduce the ability to extract features and train the network, even though it is becoming more profound than other architectures [54]. The ResNet model achieves an advanced performance in image classification relative to other models [55]. The residual mapping and shortcut connections of ResNet produce better outcomes than those of intense plain networks, and training is also more accessible [31]. ResNeSt represents a structure modified from ResNet [56], and can obtain better (or nearly better) results than ResNet.

Unlike conventional methods, the present study successfully extracted the scar mask prediction from the Mask RCNN, confirming that scar area assessment using Mask RCNN is more effective than approaches in previous studies [57]. One limitation of the conventional method is its need to remove the non-scar area to calculate the entire scar area in the dermis; thus, it takes time to calculate the scar area in the WSI. In contrast, the scar area was properly quantified using Mask RCNN during the validation process, and took less than a minute to analyze 74 images in total (including model loading and all other processing times). In addition, the results from the scar measurements in both approaches show that the Mask RCNN can measure the scar area more precisely than the conventional method, based on the standard deviation (which determines the stabilization of the data distribution).

HE staining has been widely used as a conventional tissue technique for histopathological analysis. Different tissue types often have ambiguous boundaries in the stained sections under a microscope, compared to high-precision cell imaging and fluorescence imaging techniques [58]. The K-means clustering algorithm successfully characterized the HE-stained tissue using the FS and CAS for various image sizes (Figure 8). A previous study reported that K-means clustering could be used to quantify collagen changes in chronologically aged skin by using Herovici’s polychrome stain to investigate the collagen dynamics and differentiation of the collagen by age [59]. The current study was able to analyze the changes in collagen patterns and the absence of FS as the major histological features of scar tissue. The decrease in the level of FS in the scar tissue was an indication of irreversible tissue damage, as the current findings showed a 38% decrease in the FS from the scar tissue relative to normal tissue.

The collagen density and DV of the collagen are essential features in distinguishing between normal and scar tissues. To analyze the quality of the collagen in regards to the fiber density and fiber orientation, the current research provided the CAS as the second result of K-means clustering. The CAS was analyzed by using the CDM and DV to determine both the collagen density and fiber orientation in both normal and scar tissues. The automated pixel-wise fiber orientation analysis within the histology images showed an increase in the collagen fiber alignment and collagen density during scar formation after eschar detachment. An increase in the amount of CDM in the scar tissue indicates that the inflammatory process produces additional collagen to restore the damage caused by the injury. In addition, the collagen orientation was disorganized into cellular tissues throughout the wound in a random distribution. The decreased DV of the collagen in the scar tissue caused a temporary loss of tissue elasticity during the inflammatory process.

Although rodent tissue has different tissue components and structures from human tissue, the current study suggests the feasibility of a histopathological analysis assisted by machine learning. The proposed method was able to classify and characterize the collagen structures representing the main features of scar tissue. However, limitations to the proposed method remain. Real clinical data are still needed to improve the model performance for clinical translations. The training time should be reduced further by optimizing the hyperparameters for the proposed model. The model should be robust for the skewed dataset, as clinical data tends to be sparse and imbalanced. Several methods can be to overcome the current problems, e.g., by changing the training data to reduce data imbalance, or by modifying the learning or decision-making process models to increase the sensitivity to minority classes. Some of these methods can also be grouped into data-level, algorithm-level, and hybrid approaches.

Scar tissue remodeling monitoring in clinical application is crucial for better understanding the wound healing process and evaluating treatment alternatives. The findings of this study reveal that it is possible to recognize and characterize scar tissue in a short time. Because of the modest input requirements of the model, the characteristics derived by the proposed machine learning model may result in pathologists performing less labor-intensive analysis of histological images. Collagen structures can be utilized as imaging biomarkers in various pathological research, including cancer, aging, and wound healing. By combining data from multiple stages of scar development, it is feasible to establish an objective and quantifiable histological database for research purposes. Thus, the information can aid in planning post-injury treatment that is personalized to the patient’s specific circumstances. Therefore, the proposed method in this work, which assesses the recognition and characterization of scar tissue, may prove to be a useful diagnostic tool.

## 5. Conclusions

The proposed object detection model, the Mask RCNN, accurately detects scar lesions in the WSI of HE-stained tissue. ResNet 101 is superior (as a backbone) to RPN for extracting features from HE-stained tissues. The detected ROI fed to the K-means image clustering can be used to automatically separate the structure and characterize the density and orientation of the collagen fibers. Further research will attempt to improve the Mask RCNN model to proceed with various datasets, such as with different staining methods from rat skin or clinical data.

## Figures and Tables

**Figure 1 diagnostics-12-00534-f001:**
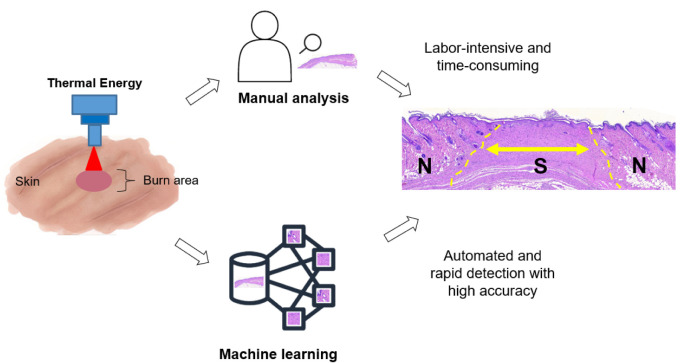
Schematic comparison between manual detection and machine learning for histology analysis of scar tissue (ML = Machine learning; N = Normal; S = Scar).

**Figure 2 diagnostics-12-00534-f002:**
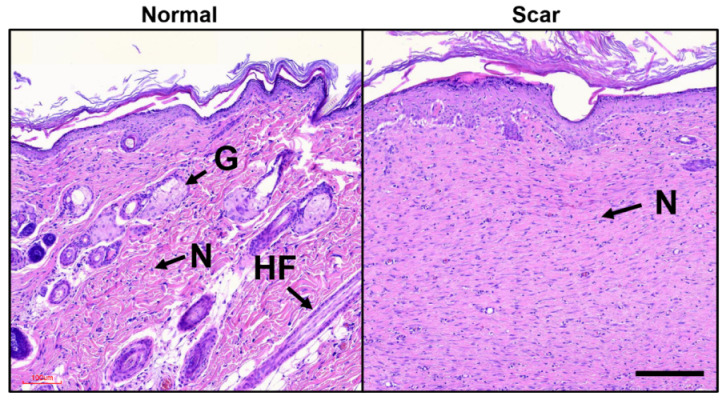
Morphology of skin tissue stained with hematoxylin and eosin (HE): normal tissue (**left**) shows hair follicles (HFs), gland (Gs), and nuclei (N) whereas scar tissue (**right**) yields the absence of both HFs and Gs (scale bar = 100 µm; 10×).

**Figure 3 diagnostics-12-00534-f003:**
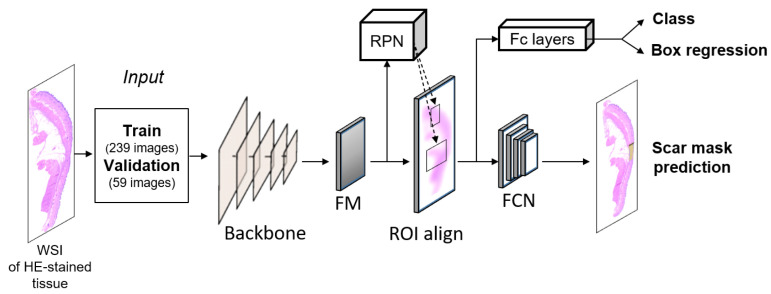
Block diagram of Mask region-based convolutional neural network (RCNN) algorithm for scar identification from histology slides (FM = Feature maps, RPN = region proposal network; FCN = fully connected network; FC layer = fully connected layer).

**Figure 4 diagnostics-12-00534-f004:**
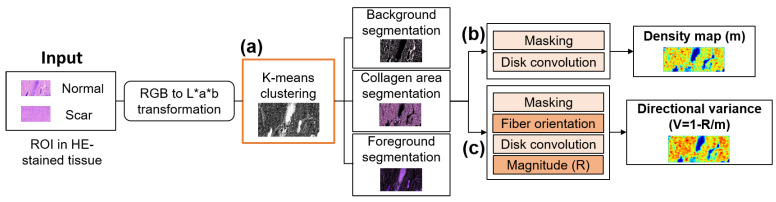
Block diagram of collagen density extractor using (**a**) K-Means clustering, (**b**) collagen density extractor, and (**c**) directional variance of collagen.

**Figure 5 diagnostics-12-00534-f005:**
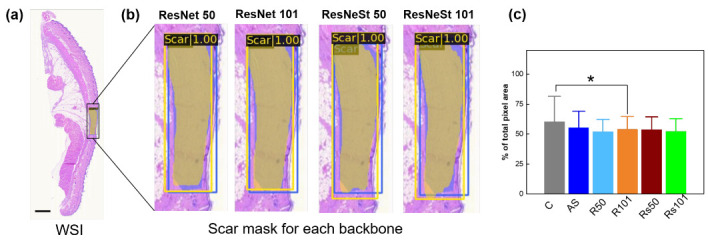
Scar identification with Mask RCNN using various backbones: (**a**) whole slide image with scar mask in yellow color and (**b**) magnified images of scar area, and (**c**) statistical comparison of measured scar areas (in pixel numbers) between conventional and machine learning methods. Blue and red boxes in (**a**,**b**) represent annotated answers (blue) and the predicted scar areas (yellow), respectively (scale bar = 3000 µm; (**c**) C = conventional method; AS = annotated scar; R50 = ResNet 50; R101 = ResNet 101; Rs50 = ResNeSt 50; Rs101 = ResNeSt 101; * *p* < 0.05 C vs. R101).

**Figure 6 diagnostics-12-00534-f006:**
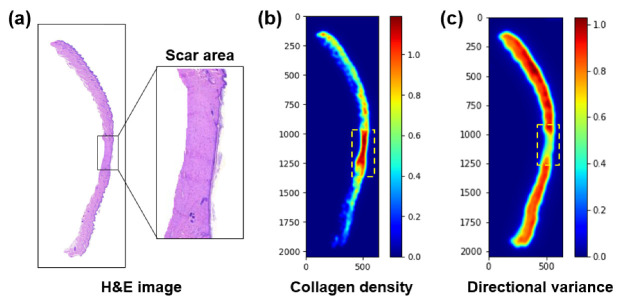
Tissue characterization from whole histology slide image: (**a**) HE-stained tissue (dermis area), (**b**) corresponding collagen density map (CDM), and (**c**) directional variance map. Yellow dashed lines represent the corresponding scar area.

**Figure 7 diagnostics-12-00534-f007:**
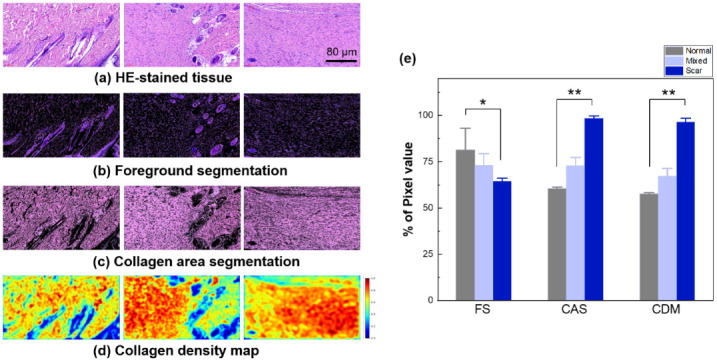
Tissue analysis with color segmentation with K-means clustering: (**a**) original dataset (HE-stained tissue), (**b**) foreground segmentation (FS; HFs, Gs, and N), (**c**) collagen area segmentation (CAS), (**d**) CDM, and (**e**) statistical comparison of FS, CAS, and CDM (input size = 886 × 1614; scale bar = 80 µm; 16X; * *p* < 0.05 and ** *p* < 0.005 between normal and scar).

**Figure 8 diagnostics-12-00534-f008:**
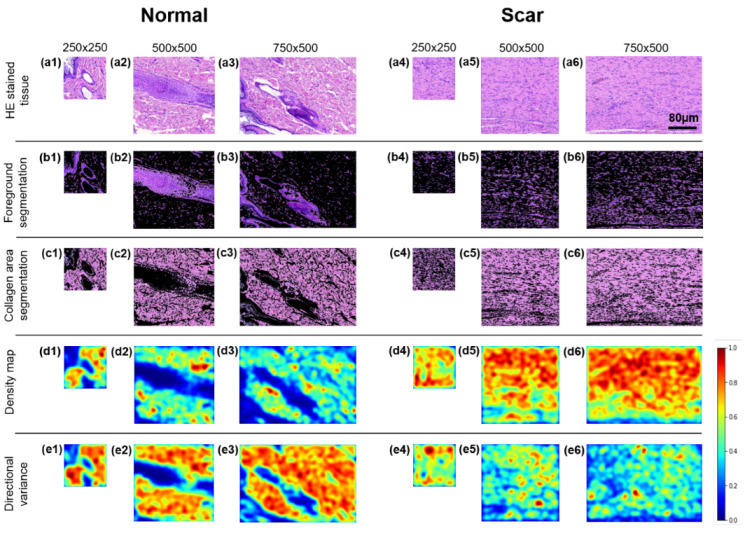
Region of interest (ROI) image comparison between normal (left) and scar tissues (right) in different image sizes (250 × 250, 500 × 500, and 750 × 500 pixels): (**a**) HE-stained tissue (a1–a3 for normal and a4–a6 for scar), (**b**) FS (b1–b6), (**c**) CAS (c1–c6), (**d**) CDM (d1–d6), and (**e**) directional variance map (e1–e6).

**Figure 9 diagnostics-12-00534-f009:**
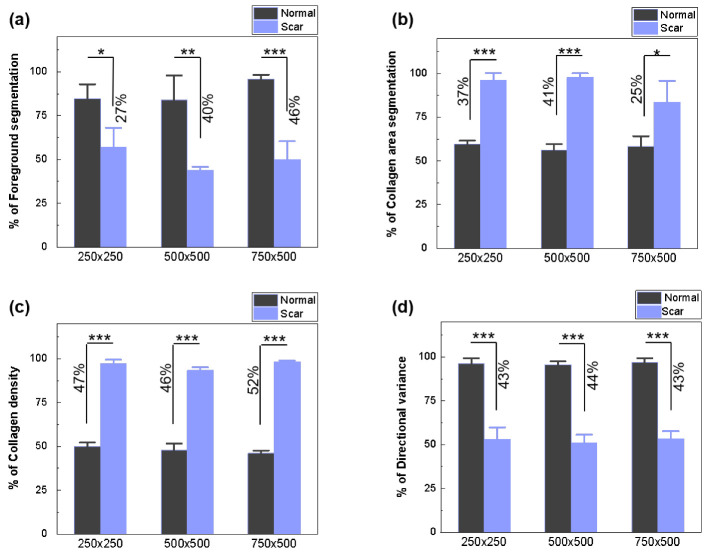
Statistical comparison between normal and scar tissues: (**a**) FS, (**b**) CAS, (**c**) CDM, and (**d**) directional variance of collagen (* *p* < 0.05, ** *p* < 0.01, *** *p* < 0.001).

**Table 1 diagnostics-12-00534-t001:** Hyperparameter setting.

Hyperparameter	Configuration
Optimizer	Stochastic gradient descent (SGD)
Learning rate	0.0025
Epoch	600
batch size	2

**Table 2 diagnostics-12-00534-t002:** Detection and segmentation results of scar by using Mask region-based convolutional neural network (R-CNN) with feature proposal network (FPN) and various backbones (ResNet50, ResNet10, ResNeSt50, and ResNeSt101): mean average precision (mAP) and mean average recall (mAR).

Backbone	${\mathrm{mAP}}^{\mathrm{bbox}}$	${\mathrm{mAR}}^{\mathrm{bbox}}$	${\mathrm{mAP}}^{\mathrm{mask}}$	${\mathrm{mAR}}^{\mathrm{mask}}$	Time (s)
ResNet 50	0.598	0.666	0.619	0.672	0.05
ResNet 101	0.620	0.680	0.631	0.677	0.07
ResNeSt 50	0.564	0.641	0.613	0.659	0.07
ResNest 101	0.597	0.672	0.587	0.645	0.09

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
