# Peer review of "Automated Structural Analysis and Quantitative Characterization of Scar Tissue Using Machine Learning"

_diagnostics, 2022, doi:10.3390/diagnostics12020534_

Round 1
Reviewer 1 Report
- In the introduction a clear discription of the problems and the benefit of the study to solve These Problems should be made.
- The paper needs language editing. It is hard to read
- What is the real clinical impact of this paper? This has to be stated and discussed!
Author Response
Dear Reviewer,
Thank you for the comments.
Please find the attached file for the response including the language editor certificate.
Sincerely,
Authors.

Reviewer 2 Report
manuscript present applying existing techniques of ML to scare tissues. Application is novel however i would suggest following changes.
- Validate methods are not described?
- where are AUC ROC confusion matrix for validation?
- In literature, grad cam like XAI methods are used to help users to understand predictions. Also progressive resizing is used to effecttively train the network. authors are advised to use them ref to https://peerj.com/articles/cs-348/.
- Data samples are less . Authors are adivsed to use GAN to augment the date for eg refer https://link.springer.com/article/10.1007/s00500-019-04602-2
- Authors have used R-CNN and related methods for predictions. New ttechnqiue which improves it is found here. Do cite and ref to this technique too. https://www.mdpi.com/1999-5903/13/12/307 andhttps://www.mdpi.com/2504-2289/6/1/9
Author Response
Dear Reviewer,
Thank you for the comments.
Please find the attached file for the response.
Sincerely,
Authors.

Round 2
Reviewer 2 Report
Good almost all suggested changes are done but not all. However, i would like to recommened accept.